# Behavior and Capacity of Moment-Frame Members and Connections during Fire

**Supriya N. Chinivar [1] and Kadir C. Sener [2,\*]**

1    Uzun + Case, LLC, Atlanta, GA 30309, USA
2    Civil and Environmental Engineering, Auburn University, Auburn, AL 36830, USA
\*    Correspondence: sener@auburn.edu; Tel.: +1-334-844-6268

**Abstract:** This paper focuses on investigating the structural behavior of members and connections that are part of moment frames under the combined effect of bending moment and thermally induced axial force during a compartment-fire event. The finite-element analysis method was employed to conduct this study using models benchmarked against experimental data from several past studies while utilizing temperature-dependent material models. A numerical parametric study on typical floor beams with slender elements for compression were conducted under combined bending and axial loading to develop interaction-capacity curves at temperatures representing fire events. The results were compared against the member-strength equations provided in the AISC Specification. The analysis results demonstrated that the AISC beam-column strength equations including the combined effects of axial-load and bending moment provided reasonable estimates for member-slenderness ratios greater than 60, but overestimated the strength of beams with slender elements for low member-slenderness ratios. Combined-load-strength studies were also conducted on a typical connection used in moment frames. The moment-connection behavior was governed by failure modes exhibited at the ends of floor beams. Therefore, the interaction equations available for beam columns resulted in conservative estimates and are recommended for calculating moment-connection capacity during compartment-fire scenarios.

**Keywords:** fire response; compartment fire; beam-column connections; combined moment and axial force; thermally induced demands; finite-element analysis; steel frame; moment connection





## 1. Introduction

Fire is a hazardous event that subjects the structure to a distinctive set of demands. In the event of a fire hazard, the structure's resistance is dependent on its ability to resist potential load demands, such as gravity loads, lateral loads, and induced thermal loads, and requires satisfactory resilient performance under fire demands. Available code-based design approaches rely on load combinations where envelope demands from one hazard type typically govern the design, thus preventing the interaction of hazard scenarios. However, it has been seen from recent events that multi-hazard scenarios can have catastrophic consequences on the structural performance at the system level; for example, having a compartment fire after a major seismic event, as observed during the 1994 Northridge and 1995 Kobe earthquakes, which subsequently led to various fire events [1]. More recent post-earthquake fire events have also been observed following several earthquakes in Japan [2] and the United States [3].

Recent studies aimed at improving the fire resistance of steel structures using the prescriptive-method focus on innovating new materials and methods to protect the structural members by delaying the onset of critical-failure temperature, thereby enhancing the structural performance (e.g., [4]). These fire-protection materials may prove to be advantageous in delaying the temperature rise of structural members as a prescriptive measure; however, developing structural fire-engineering methods through guidelines for

analysis and design of structures to withstand fire events are expected to be more efficient and economical solutions [5].

Structural integrity of beam-to-column connections is essential during extreme events in ambient or elevated temperatures, as connection failures may lead to premature system failures regardless of the condition of the joining members. These connections are generally designed as simple (shear) connections if used as part of the gravity frame in the case of the structure having a separate lateral load-resisting system. Simple connections experience marginal levels of moments at ambient conditions by allowing large rotations [6]. These deformations are more prominent in fire situations, and the forces generated through the floor beams are transferred to the girder or columns via connections. Due to the relative positioning and limited exposure, connections tend to heat up more slowly than the sections within the span, reducing the temperature-dependent softening within the connection [6]. Fire tests conducted on the Cardington full-scale test frame [7] and observations from actual fire events [8] have illustrated the importance of studying the connection behavior at elevated temperatures, as the influence of connection rigidity has a more substantial impact on increasing the survival time of the structure [8] as compared to the structural members. Recent experimental research by the National Institute of Standards and Technology (NIST) on simple connections [9] demonstrated that although these connections are flexible and allow expansion of the beam, they experienced significant stress levels near the beam capacity due to combined thermally induced axial forces and bending-moment demands. This outcome might have concerning implications for moment connections, which are stiffer than simple connections, and the thermally induced force and moment demands generated at the beam ends are expected to be greater relative to the respective member capacities. The higher demands are likely to increase the potential of leading to local failures and instabilities (rupture, buckling) within the member or the connection.

Computational studies have also been performed to investigate the behavior and response of frames with moment connections under fire loading and various restraining conditions. NIST researchers [10] analyzed multi-bay frames from a prototype building representing typical steel-composite construction in the US [11] while simulating the heating and cooling phases of compartment fires. The studies consisted of analysis of two types of moment frames from the prototype building that included an intermediate moment frame with welded unreinforced flange-bolted web (WUF-B) connections and a special moment frame with reduced beam-section (RBS) connections. The moment frames investigated in the study were portions of the prototype building. Detailed finite-element analyses of the moment-frame assembly included different restraint-level assumptions (full, partial, and unrestrained) provided by the adjacent frames. The assembly was subjected to a uniform gravity load along the beam, and the axial load was measured at beam ends with uniformly changing beam temperatures. The computational results indicated extensive yielding and damage in the connections due to the heating- and cooling-temperature reversal representing fire at the gravity-load condition. The frame behavior was highly dependent on the restraint condition of the joint from the adjacent bay, as Figure 1 presents the normalized axial-load response plotted against the applied temperature for a W21x73 member with a WUF-B connection. The figure indicates that the thermally induced axial-force demands reached about 80% of the axial capacity of the beam (compared to $A_g \cdot F_y(T)$, where $F_y(T)$ is obtained from Appendix 4 of AISC Specification [12]). This observation was valid regardless of the restraint level of the adjacent bay, but the forces nearing member capacity were achieved at different temperature instances for each restraint level. The response of the unrestrained and partially restrained cases differed drastically from the fully restrained case, as the former cases failed due to shear rupturing of bolts. The fully restrained beams failed due to local buckling followed by bolt-shear rupture in the heating phase, with partial fracture of the flanges resulting from tensile forces developed in the cooling phase.

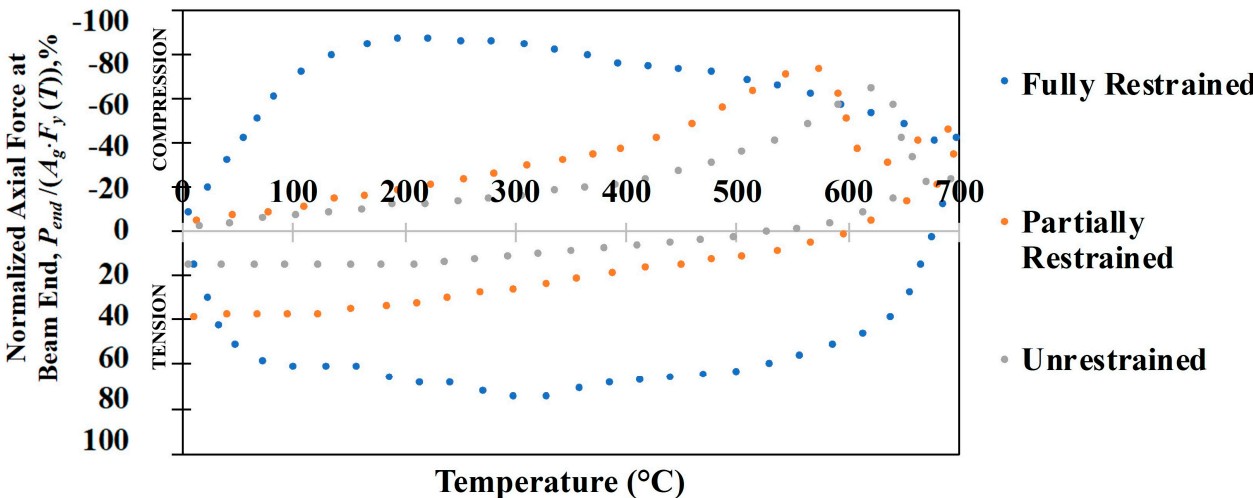

**Figure 1.** Normalized axial-end force variation during fire-temperature cycle for W21x73 with WUF-B connection, reproduced from [10].

A literature review on the topic revealed a lack of extensive experimental or numerical studies on steel moment-frame members and connections subjected to fire, where the connection is expected to experience combined axial force-bending moment interaction at elevated temperatures. A few studies were performed at elevated temperature conditions where it was confirmed that the axial force affects the connection structural behavior substantially, primarily moment connections including extended end-plate connections [13]. From the above discussion, it is clear that the effect of the axial force on the behavior of joints at elevated temperatures is not fully understood and the strength of steel moment connections has been overlooked, which may result in overestimating the fire resistance. Therefore, additional research is necessary to better understand the extent of axial-restraint influence on the connection response and establish performance-based design methods for fire safety.

## 2. Objective and Approach

The objective of the presented study was to investigate the structural behavior and capacity of members and connection of moment frames subjected to combined effects of bending-moment and thermally induced axial-force demands at elevated temperatures through computational evaluations. A member-level study of floor beams, typically classified to have slender elements for compressive loading, was conducted through finite-element analysis to develop combined axial-load and bending-moment interaction equations for elevated temperatures (M-N-T). The developed M-N-T interaction relationships were compared against established code equations, such as the guidance provided for member strengths in AISC Specification Appendix 4 [12]. Numerical investigations were also conducted on moment frames using welded-flange and bolted-web connections (WUF-B), as a representative moment connection, to develop similar moment–axial force–temperature (M-N-T) interaction curves for moment connections.

Detailed three-dimensional nonlinear structural models were developed to predict the nonlinear inelastic behavior of steel moment connections using a commercially available finite-element software, Abaqus [14]. The temperature-dependent mechanical properties used in the computational studies for various steel members were obtained both by studies by the NIST researchers [15] and the European Committee for Standardization (CEN or Eurocode) publications [16] to investigate the sensitivity of the results. Once the M-N-T interaction curves for the moment connections considered for the study were established, they were compared against the M-N-T interaction curves developed using the member-strength equations given in the AISC Specification.

The developed models were initially evaluated and benchmarked against the experimental beam-column sub-assembly tests conducted by previous researchers, which include Yang et al. [17], who studied welded flange-bolted web-type moment connections, and Al-Jabri et al. [18,19], who studied end-plate moment connections. The benchmarking process of the finite-element-modeling methodology was performed by comparing load versus displacement, moment versus rotation, displacement or rotation versus temperature responses, and failure modes observed during the experiments.

These verified models were further used to conduct a parametric study on the behavior of floor-beam members at various combined load and temperature scenarios. A member-level study on typical beam sections for elevated temperatures representing fire was conducted. The study involved the assessment of the (1) compressive strength of the member (compression-only case), (2) flexural strength of the member (flexure-only case), and (3) combined axial-load and bending-moment capacity (M-N-T). The fundamental behavior of beam-column moment connections was also investigated using the WUF-B as a commonly used moment connection while subjecting to axial, moment, and combined force cases at elevated temperatures to establish M-N-T relationships. The influence of materials models and element fracture on the analysis results was also studied.

## 3. FEM Model Development and Benchmarking

A detailed three-dimensional, nonlinear, stress-analysis model was developed in Abaqus using the built-in constitutive material models available in the software [14]. The analysis was conducted using a dynamic explicit numerical-solution technique and utilized temperature-dependent material models for steel to predict the nonlinear and inelastic behavior of steel-moment connections and members subjected to mechanical loads at elevated temperatures. These temperature-dependent material properties were incorporated using the methodology given by previous researchers [20,21], which included defining nonlinear mechanical stress–strain–temperature ($\sigma$-$\varepsilon$-T) relationships for steel.

The steel beam-column and their connections were modeled using the first-order eight-node linear brick elements with reduced integration and hourglass control (C3D8R). For the bolted connections, the bolt holes were considered 2 mm (~1/16 inch) larger than the bolt-shank diameter, and the bolt threads were assumed to be excluded from the shear planes. The bolted connections were modeled using surface-to-surface contact definitions incorporating a finite-sliding option between the contacting surfaces of the connection. As the contact formulation identifies the surfaces in contact, penalty-type contact interactions captured the bolt-bearing behavior of the contacting surfaces throughout the analysis. The bolts were assumed to be snug-tight using standard high-strength bolts; therefore, bolt pre-tensioning was not considered in the models. A friction coefficient of 0.3 was assigned for all the contacting surfaces according to the recommendations in the AISC Specification for Class A surfaces [12]. The weld metals used in the connections were not explicitly modeled and were represented using a tie constraint between the connecting elements due to welds having higher strength compared to plates or rolled steel shapes.

To achieve high accuracy in the contact interactions between the connecting parts, fine meshing in the connection regions was implemented in the simulations. The meshed boundaries of common structural elements (beam member, shear tab, and high-strength bolt) used in the computational models is illustrated in Figure 2. Symmetry-boundary conditions were assigned at the end of members to achieve representative subassembly behavior in the models.

### 3.1. Temperature-Dependent Material Models for Steel

Several material models for the mechanical properties of steel members at elevated temperatures are available in the literature. One of the most common models is provided in Eurocode 3 [16], which contains provisions for analyzing structural steel and concrete members at elevated temperatures. Eurocode 3 considers rate-dependent effects such as thermal creep while defining the stress–strain relationships at various temperatures. The

Eurocode material models are also the basis for the tables provided in AISC Specification. The table in the Specification uses reduction factors for the elastic modulus, proportional limit, and yield strength ($k_E$, $k_p$, and $k_y$, respectively), and these reduction factors are summarized in Figure 3.

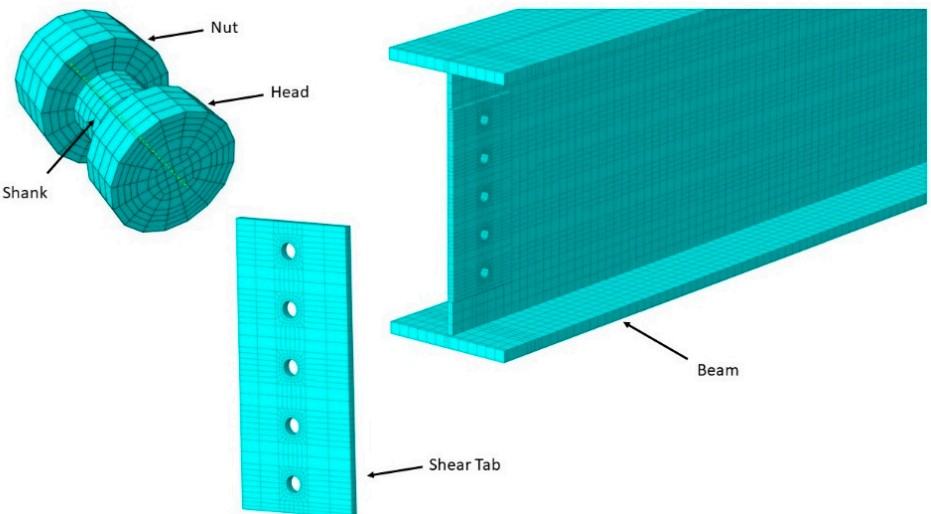

**Figure 2.** Mesh patterns for bolts, shear tabs, and members used in FEM models.

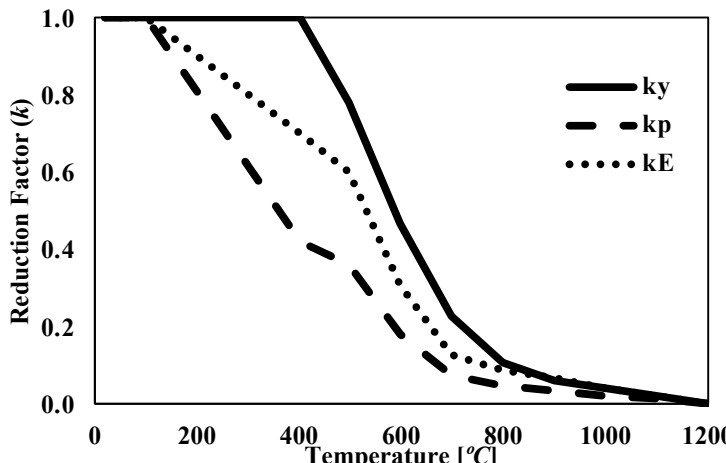

**Figure 3.** Reduction factors for the stress–strain relationship of carbon steel at elevated temperatures used in the AISC Specification [12] and Eurocode 3 [16].

NIST researchers have also developed uniaxial stress–strain curves at various temperature levels (σ-ε-T) for structural steel types commonly used in US building construction [15]. These guidelines were recently published in a report on fire-resistant structural design of concrete and steel buildings [22] and recommended the use of newly developed σ-ε-T relationships for different types of steel including plates and bolts. The NIST guidelines to account for elevated-temperature effects are based on reduction factors similar to the AISC tables.

Figure 4 compares the Eurocode 3 [16] and NIST [15] true stress–strain relationships at 200 °C, 400 °C, and 600 °C. Since the NIST σ-ε-T relationships do not account for rate-dependent effects such as thermal creep, higher yield stress, and greater post-yield strain-hardening responses were observed compared to the corresponding Eurocode 3 σ-ε-T relationships. Another fundamental difference between the models is that the reduction factors defined in the NIST equations were calculated based on the 0.2% offset strain, whereas Eurocode considers a 2% offset strain, which results in relatively low stiffness and strength properties. Other researchers (e.g., [23]) have also made similar observations

when comparing the NIST and Eurocode 3 material models in their studies, wherein the Eurocode 3 results in conservative predictions on the structural behavior and response due to the lower mechanical-property assumptions relative to those of the NIST material model. For example, the mechanical properties given according to Eurocode 3 begin to soften when temperatures exceed 100 °C, as shown in Figure 3. When compared for another temperature case, at 600 °C, the Eurocode 3 elastic modulus is about 52% of the NIST value.

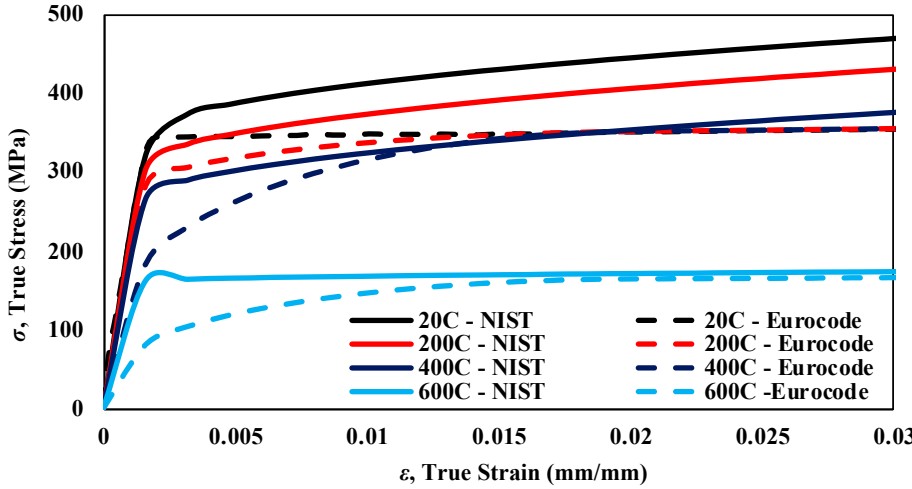

**Figure 4.** Comparison of stress–strain–temperature curves provided by Eurocode [16] and NIST [15].

*3.2. Benchmarking of FEM Models*

The FEM methodology was benchmarked against two experimental-test programs that focused on moment frames and verified by comparing the analysis results.

3.2.1. Sub-Frame Tests by Yang et al. (2009)

In the first study, the experiments by Yang et al. (2009) [17] included steady-state and transient-temperature conditions to examine the stiffness- and strength-degradation effects on moment connections. The test program was conducted on sub-frame assemblies of a commonly used moment connection. The structural response of the sub-frame was investigated in the complete loading regime to observe global and local collapse when subjected to temperatures simulating fire conditions. The experimental setup used in this research included an H600 × 300 × 12 × 25 mm beam, and an H600 × 600 × 25 × 36 mm column to form the sub-frame as a picture of the test specimen is shown in Figure 5a. The beam was 3100 mm in length, and the loading was applied at 2900 mm from the column flange with lateral supports provided at 2000 mm from the column flange to prevent lateral-torsional buckling (LTB) of the steel beam. The testing was conducted in a large furnace that enclosed the sub-frame specimen.

The researchers completed four sub-frame tests with WUF-B connections subjected to bending moments at temperatures representing compartment-fire conditions. Two steady-state tests were performed by uniformly heating the specimens to specified temperatures of 550 °C (Specimen 1) and 650 °C (Specimen 2) and followed by gradually increasing load until failure. On the other hand, in the transient testing phase, service-level loads applied to the identical steel column-beam subassemblies were kept constant and followed by heating the specimen. The transient-loading protocol was to determine the temperature resistance of the structure under sustained loading. Two transient-temperature tests were performed with one specimen covered with three-hour fire protection, and therefore is not analyzed in this study. The unprotected subassembly test (Specimen 3) was conducted by increasing the furnace temperature up to 1100 °C by following the standard temperature–time curve provided in ISO Standard 834 [24] after the sustained-load level was reached. Parts of the subassembly members reached up to 770 °C during the transient-temperature tests. The

results from the tests conducted at steady-state conditions indicated up to a 60% stiffness reduction and 40% strength degradation for the temperatures considered in the study. For the transient-heating case, the member deformations increased rapidly once the connection reached temperatures beyond 550 °C. The failure modes were concentrated near the welded flanges in all the tests, as the resistance provided by the web bolts was reported to degrade due to the loss of pre-tension and observed slippage.

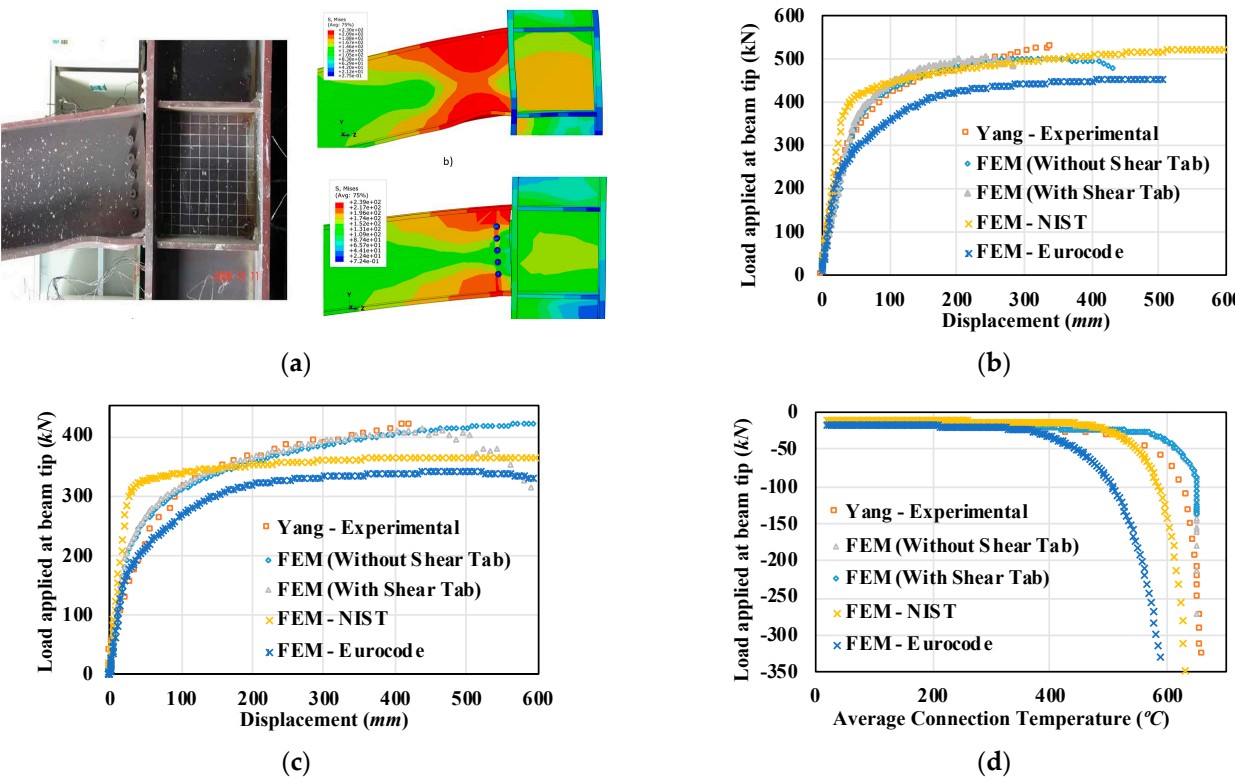

**Figure 5.** Comparison of tests by Yang et al. [17] with FEM results: (**a**) deformed shape and local buckling for Specimen 2 (photo reused with permission from Ref. [17]); (**b**) load deflection for Specimen 1 (heated to 550 °C); (**c**) load deflection for Specimen 2 (heated to 650 °C); (**d**) load deflection for Specimen 3 (transient-load protocol).

The FEM methodology summarized in the previous section was used to develop models and analyze previously tested subassemblies. The material properties of the steel members used in the analysis were obtained based on the uniaxial-tension tests conducted by the authors at various temperature levels observed during the tests [25]. The temperature-dependent mechanical properties provided in NIST and Eurocode 3 were also used to observe their influence on the results. Two types of models were developed, where the first model included the shear-tab and bolted connection, and the other model omitted the shear-tab connection by having the beam fully tied to the column. Figure 5a visually compares the failure modes for Specimen 2 for the models with and without the shear tab against the experimental response near the connection region, where local flange buckling observed during the test was captured in the models.

The FEM analysis results included four responses based on the (i) measured material properties by the authors [25] with the shear tab, (ii) measured material properties by the authors [25] without the shear tab, (iii) temperature-dependent properties according to NIST [15] without the shear tab, and (iv) temperature-dependent properties according to Eurocode 3 [16] without the shear tab. Figure 5b,c present the load- vs. displacement-response comparisons between the results obtained through the experimental and analysis results for the steady-state tests at 550 °C (Specimen 1) and 650 °C (Specimen 2), respectively.

The initial stiffness (secant stiffness at 100 kN) was within 3% and 2% of the experimentally measured values when the measured properties were used, 25% and 31% for the NIST model, and 14% and 22% for the Eurocode model for Specimens 1 and 2, respectively. The peak forces were within 4% and 6% of the experimentally measured strengths when the measured properties were used, 1% and 7% for the NIST model, and 14% and 19% when the Eurocode model was used for Specimens 1 and 2, respectively. The failure temperatures were within 2%, 5%, and 12% for the measured, NIST, and Eurocode models when used for Specimen 3, respectively, indicating improved accuracy when the measured properties provided by the authors were used. The FEM results were in close agreement with the experimental results regardless of including the shear-tab connection (within 2% difference) when using the reported material properties; therefore, the material-model study was conducted only on the fully tied model due to the simplicity.

For Specimen 1, which reached 550 °C during the test, the ultimate peak load was reported as 508 kN, compared to the peak load obtained from the FEM analysis using the measured properties with or without the shear tab, which was 502 kN. Similarly, for Specimen 2, which reached 650 °C during the test, the experimental peak load was reported as 420 kN, whereas the FEM analysis using the measured properties with or without the shear tab predicted 422 kN. The comparisons were less favorable for the cases that used the material properties according to NIST and Eurocode 3, with the Eurocode models being the least accurate. The Specimen 2 results deviated particularly in the earlier part of the response but became comparable after the yield in the specimens. The analysis results using the NIST and Eurocode models showed a load-plateau behavior after yield. The plateau response was less prevalent in the experimental results, which indicated strain hardening after yield. This discrepancy was expected due to the differences in the reported measured properties by the researchers and the NIST and Eurocode 3 curves.

Figure 5d shows the comparison between the experimental and finite-element analysis results for Specimen 3, which was subjected to the transient-temperature-loading protocol. The models were analyzed for uniformly increasing temperature profile for all the subassembly portions due to the low scatter of temperature variation reported by the authors [25]. Similar to the steady-state specimens, the finite-element results showed close agreement with the experimental results when using the material properties measured by the authors. The Eurocode 3 material model produced less accurate but more conservative results compared to the NIST model, where the Eurocode 3 material model had about a 20% softer response in the load-versus-displacement plots. On the other hand, the NIST material model showed relatively good agreement against the experimental results. The comparisons verify that the Eurocode 3 material model produced lower bound results compared to the more accurate NIST material model.

### 3.2.2. Moment-Frame-Arrangement Tests by Al-Jabri et al. (2006)

Al-Jabri et al. (2006) [26] conducted experimental investigations on fully and partially rigid connections to provide moment–rotation–temperature relationships for various practical-moment connections. The testing program included full-scale beam-to-column connections comprising 20 specimens in five groups. The specimens had symmetric cruciform arrangements with 1.9 m-long cantilever beams connected to either side of a 2.7 m-high center column. The testing was conducted by subjecting the connection to a constant bending moment while increasing the furnace temperature to 900 °C in 90 min. Group 1 and Group 2 specimens with rigid end-plate connections without composite concrete decks were chosen for developing the benchmarked finite-element methodology detailed in the previous section. Similar to the previous study, the models were analyzed by uniformly increasing the subassembly temperature based on the temperature experienced for the connection region.

The test specimens of Group 1 comprised two 254x102UB22 (US equivalent of W10x15) beams connected to a 152x152UC23 column with 8 mm-thick flush end plates using six M126 Grade 8.8 bolts. In Group 2, the specimens comprised a pair of 356x171UB51 beams

connected to a 254x254UC89 (US equivalent of W14x34) column by 10 mm-thick flush end plates with eight M20 Grade 8.8 bolts. Each specimen group included four identical specimens that were tested with applied connection moments corresponding to 20, 40, 60, and 80% of the connection moment capacity, which was calculated to be 140 kNm. The test results were presented in the form of temperature-rotation plots and included discussions of the failure modes.

As there was no measured information on the temperature-dependent mechanical properties of the steel used in the members, the NIST material model was utilized for the steel members and bolts for the analysis. The NIST model was preferred over the Eurocode due to it comparing more favorably against the studies by Yang et al. [17].

The results obtained from the finite-element analysis of Group 1 and 2 specimens are compared against the experimental results in Figure 6. The finite-element results show close agreement with the experimentally obtained temperature-rotation responses in Figure 6a,b. Group 2 specimens, with thicker end plates, indicated a stiffer response than the Group 1 plate, with lesser deformation at the same temperature levels. The results in both the small-rotation (elastic region) and large-rotation (plastic deformation) regions of the responses correlated well with the experimental results. The overall response of the subassemblies was captured well by the model at a wide range of temperatures. The cruciform specimens with the special loading arrangement simulated a negative moment case in the connection as the beam's bottom flange was in compression and the top flange was in tension. The FEM results were also able to predict the local deformation of the top plate due to high tensile stresses on the top bolts. As shown in Figure 6c, the column flange deformed due to the force couple, and the analysis was able to predict the column's local web buckling around the bottom compression region of the connection.

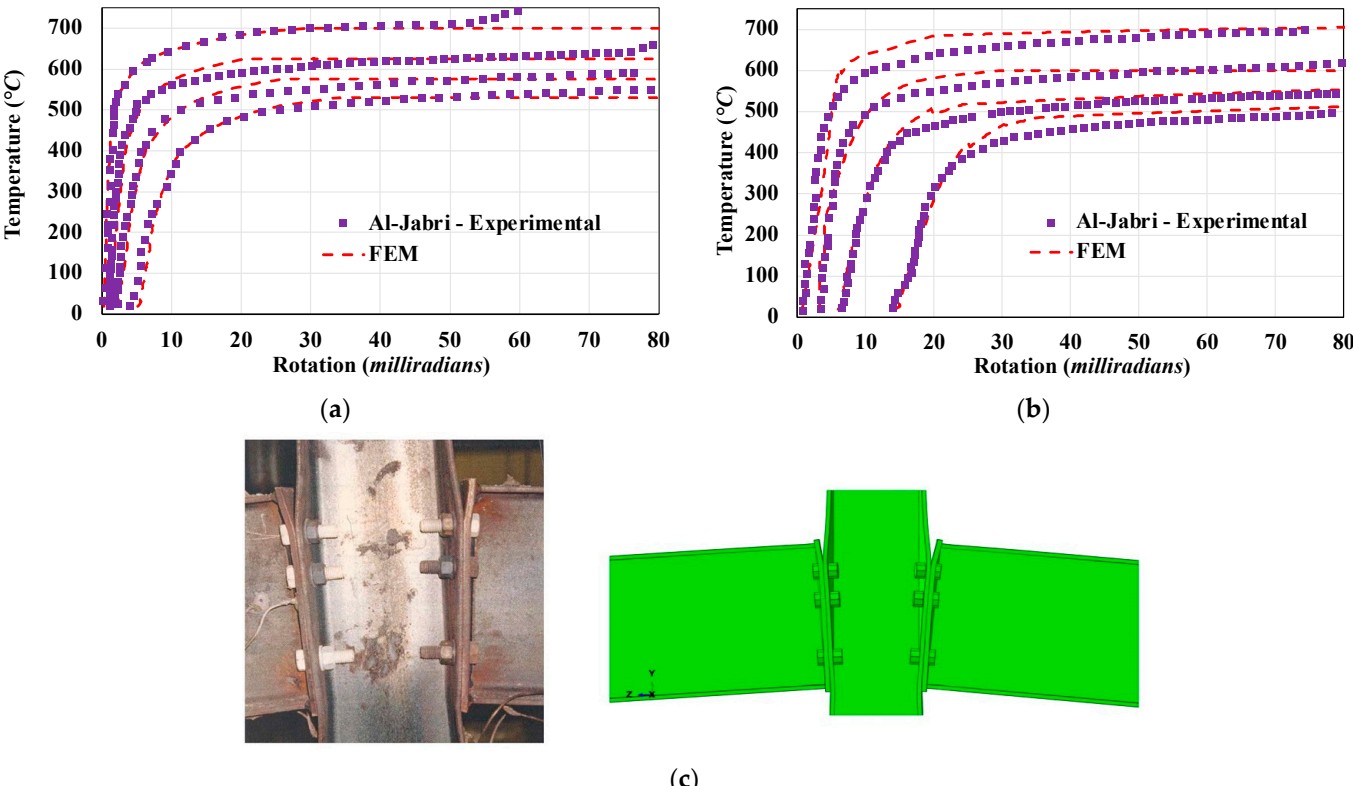

(a)

(b)

(c)

**Figure 6.** Comparison of tests by Al-Jabri et al. [18] with FEM results: (**a**) temperature rotation for Group 1 specimens; (**b**) temperature rotation for Group 2 specimens; (**c**) local flange deformations at maximum load (photo reused with permission from Ref. [18]).

## 4. Member-Strength Assessment at Elevated Temperatures

### 4.1. AISC—Appendix 4

Appendix 4 of the AISC Specification [12] stipulates design requirements for evaluating structural steel components for fire conditions. The provisions allow advanced analysis to be performed by incorporating thermal and mechanical response of the design-basis fire or using simplified methods, which involves calculating the member-based strength for individual demands such as compression and flexure. The AISC Appendix 4 equations for temperature-dependent nominal strength of individual members were developed based on the work by Takagi and Deierlein (2007) [27]. Their study originally targeted compact steel members, whereas this study evaluates their applicability for slender members, particularly beam members that have slender elements for compressive design, indicating a potential for local flange or web buckling under compression.

The temperature-dependent nominal compressive strength ($F_{cr}$) provided in Appendix 4 is shown in Equation (1) (Equation A-4-2 in AISC 360-16 [12]). The equation uses temperature-dependent yield strength, $F_y(T)$, and the elastic modulus, $E(T)$, which are obtained using the reduction factors provided in Figure 3. The equations follow a similar format as the flexural-buckling equations for columns in ambient conditions, as provided in Chapter E of the Specification. In the equation, $F_e(T)$ is the critical elastic-buckling stress accounting for the temperature effects, $L_c$ is the effective length, and $r$ is the radius of gyration of the beam on the weak axis.

$$F_{cr}(T) = [0.42^{\sqrt{(F_y(T)/F_e(T))}}]\cdot F_y(T), \text{ where } F_e(T) = (\pi^2\cdot E(T))/(L_c/r)^2 \quad (1)$$

The nominal flexural strength ($M_n$) for lateral-torsional buckling of unbraced doubly symmetric members at elevated temperature is also provided in Appendix 4 of the AISC Specification [12]. Similar to the flexural-buckling strength equations (Equation (1)), this set of equations also follows a similar format as the ambient-strength provisions given in Chapter F of the Specification and uses the temperature-modified versions of the parameters included in the equations. Inelastic lateral-torsional buckling (LTB) controls when the unbraced length, $L_b$, is less than the limiting laterally unbraced length for inelastic lateral-torsional buckling, $L_r(T)$, which is provided in Equation (2) (Equation A-4-6 in AISC 360-16). The nominal-flexural-strength equation for inelastic LTB is provided in Equation (3) (Equation A-4-3 in AISC 360-16). The parameters used in the equation are the plastic moment, $M_p(T)$, and yield moment (accounting for residual stresses), $M_r(T)$; capacities modified for elevated temperatures ($T$); and the modification factor for nonuniform moment diagrams ($C_b$). The nominal-flexural-strength equation for elastic LTB (applies when $L_b > L_r(T)$) is provided in Equation (4) (Equation A-4-4 in AISC 360-16) and is obtained by multiplying the temperature-dependent critical elastic-buckling stress, $F_{cr}(T)$, with section modulus, $S_x$. Finally, the beam-column design is checked against demands, $P_r$, $M_r$, using Equations (5) and (6) by combining the strengths obtained from individual compressive and flexural strengths of the members (Equations H1-1a and H1-1b in AISC 360-16).

$$L_r(T) = 1.95\, r_{ts}\cdot E(T)/F_L(T)\cdot\sqrt{[J/(S_x\cdot h_0) + \sqrt{((J/(S_x\cdot h_0))^2 + 6.76\cdot(F_L(T)/E(T))^2)}]} \quad (2)$$

$$M_n(T) = C_b\cdot(M_r(T) + [M_p(T) - M_r(T)]\cdot[1 - L_b/(L_r(T))]^{c_x}) \leq M_p(T), \text{ where } c_x = 0.6 + T/250 \quad (3)$$

$$M_n(T) = F_{cr}(T)\cdot S_x \leq M_p(T) \quad (4)$$

$$P_r/P_c + (8/9)\cdot M_r/M_c \leq 1.0 \text{ when } P_r/P_c \geq 0.2, \quad (5)$$

$$P_r/2P_c + M_r/M_c \leq 1.0 \text{ when } P_r/P_c < 0.2. \quad (6)$$

### 4.2. Strength Assesment Using Benchmarked Numerical-Modeling Approach

The same beam sections studied by Takagi and Deierlein [27], i.e., W14x90 and W14x22 (ASTM A992), were used in the analysis for investigating member strengths for individual and combined loading cases. In addition, a W21x73 section was also considered in the

analysis to include a commonly used beam size and for use in a subsequent numerical parametric study. The W14x22 and W21x73 steel sections are both compact for flexure but slender for compression design, according to the Specification. Conversely, the W14x90 steel section is non-slender for compression but non-compact for flexure. The strength assessment of these steel sections at elevated temperatures was to evaluate the (i) compressive strength of members (compression-only case investigating flexural buckling), (ii) flexural strength of laterally unsupported steel members (flexure-only case investigating LTB), and (iii) capacity of steel members under combined axial loads and bending moments (beam-column case).

The study by Takagi and Deierlein [27] used the Eurocode material models to account for the nonlinear stress–strain curves of steel at elevated temperatures. In order to evaluate and expand up on their results, a numerical parametric study was conducted using the same structural steel shapes and material properties based on the Eurocode 3 equations. For the compression-only case, the flexural-buckling capacity was evaluated by subjecting the members to axial loads at varying slenderness ratios ($L_c/r$). For the flexure-only case, the flexural strengths were evaluated by subjecting the members to bending moments on the major axis for a number of different member-slenderness ratios ($L_b/r$).

The nominal strength properties at ambient and elevated temperatures used in the FEM analyses were kept consistent with the original study [27] to provide a consistent comparison of the results with the design equations in Appendix 4. The temperature distribution in the members was uniform throughout the length and cross-section. In these member studies, simply-supported boundary conditions were arranged to be the same as described in the original study, where the displacements along the weak and strong axes (X- and Y-axis) and twisting rotation on the Z-axis were restrained at both ends, as shown in Figure 7. The longitudinal displacements along the Z-axis were restrained at one end, and the rotational displacements on the weak and strong axes were kept free at both ends. For the compressive-strength analysis, the axial force was applied by a kinematically restrained coupling function to ensure uniform load application at the ends of the member. For the flexural-strength analyses, a force couple in opposite directions was applied on the top and bottom flange at each end to induce a uniform moment on the strong axis.

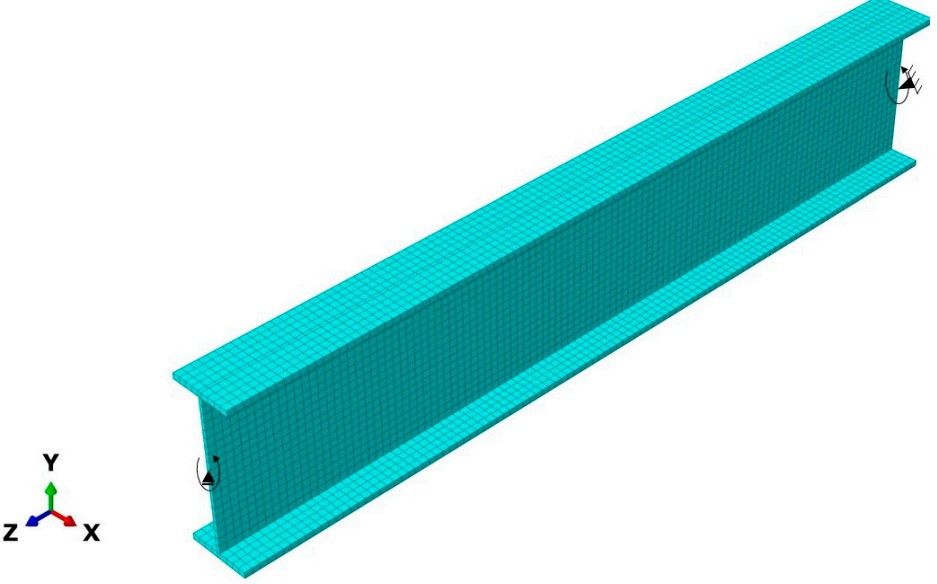

**Figure 7.** Boundary conditions used for member-strength assessment studies in FEM model.

4.2.1. Compressive-Strength Assessment

The FEM results for the compression-only case of W14x22 at 500 °C are compared to the results reported by Takagi and Deierlein [27] and the current AISC [12] equations for column strength in Figure 8. The comparisons were conducted for a wide range of slenderness ratios ranging from 20 to 200 and indicate an agreement for slenderness ratios ($L_c/r$) greater than 60. However, the finite-element results diverged from the AISC equations for low member-slenderness ratios ($L_c/r < 50$). The effect of web slenderness was clearly seen at these low slenderness ratios, as the failure became controlled by local buckling and deviated from the AISC curve.

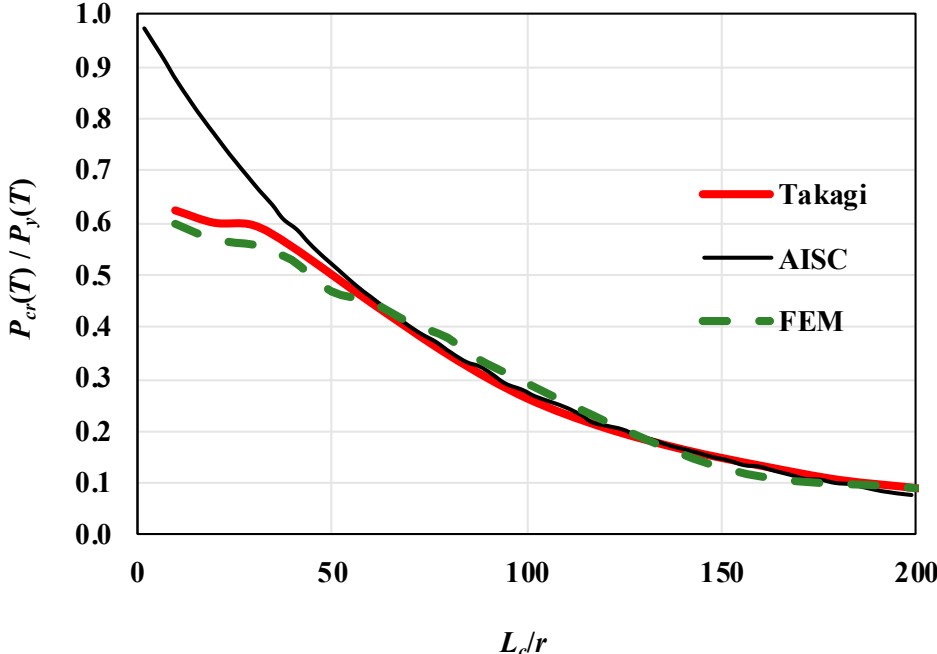

**Figure 8.** Comparison of compressive strength for W14x22 at 500 °C.

An additional compressive-strength study on a W21x73 member (not considered in the original study) was conducted at various temperature levels to reinforce the outcomes. Figure 9 shows the comparison between the finite-element results for compression-only loading at 20 °C, 200 °C, 400 °C, and 600 °C against the AISC [12] equations for varying slenderness ratios ($L_c/r$). The study confirms that the compressive-strength variation with slenderness ratios agreed with the current AISC equations for high slenderness ratios, but large discrepancies occurred for low slenderness ratios where local web buckling became prevalent. Figure 10 shows stress-contour plots for W14x22 and W21x73 sections at the slenderness of $L_c/r = 20$ at 500 °C to demonstrate the local-buckling modes observed for these cases. This local-buckling limit is not reflected in the current AISC column strength equations in Appendix 4 [12], since these member lengths are considered impractical from a realistic member-length perspective ($L_c/r = 50$ corresponds to 1.3 m for W14x22 and 2.3 m for W21x73). Thus, these equations provided unconservative results for low member slenderness, as they tended to predict compressive strength up to 40% higher than the FEM capacity.

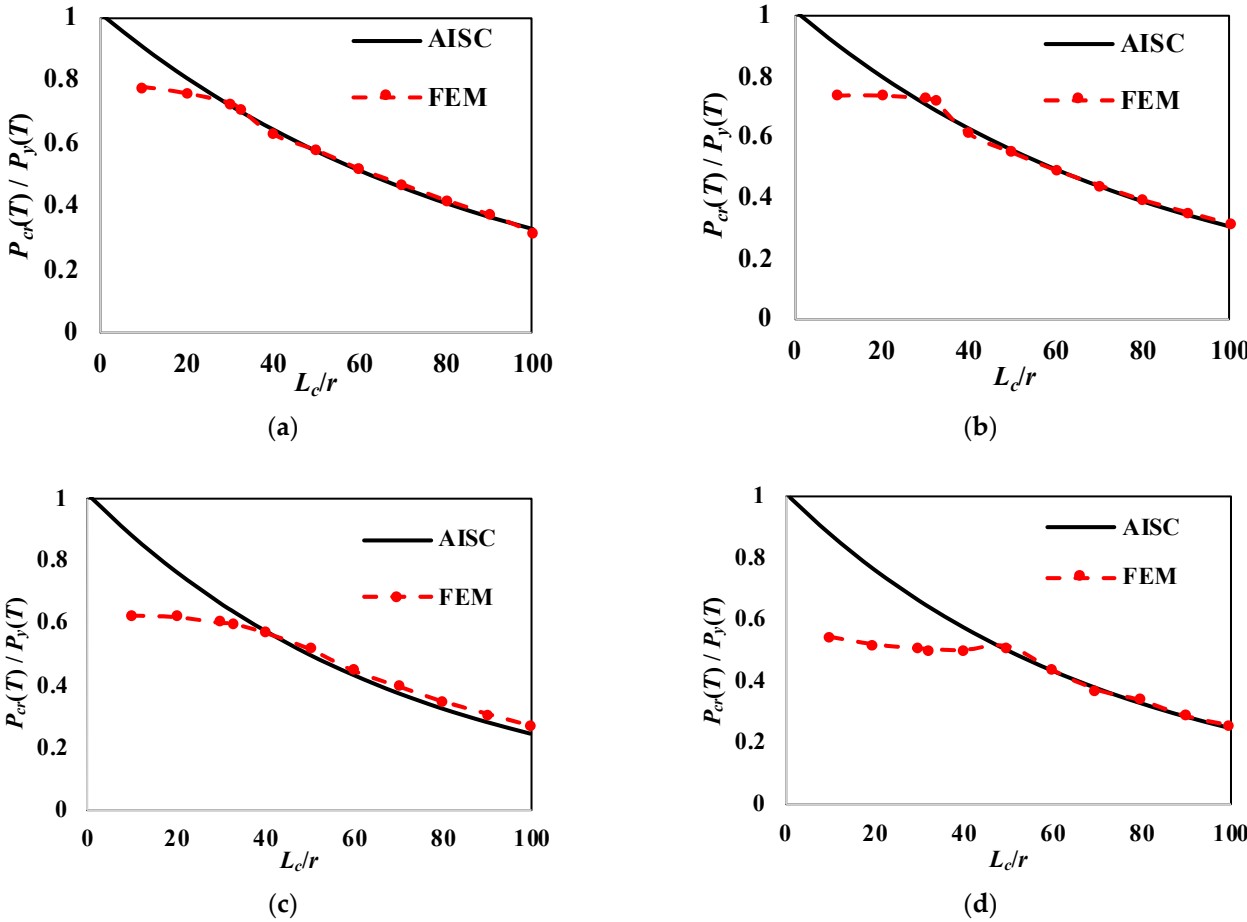

**Figure 9.** Compressive-strength curves for W21x73 at (**a**) 20 °C, (**b**) 200 °C, (**c**) 400 °C, and (**d**) 600 °C.

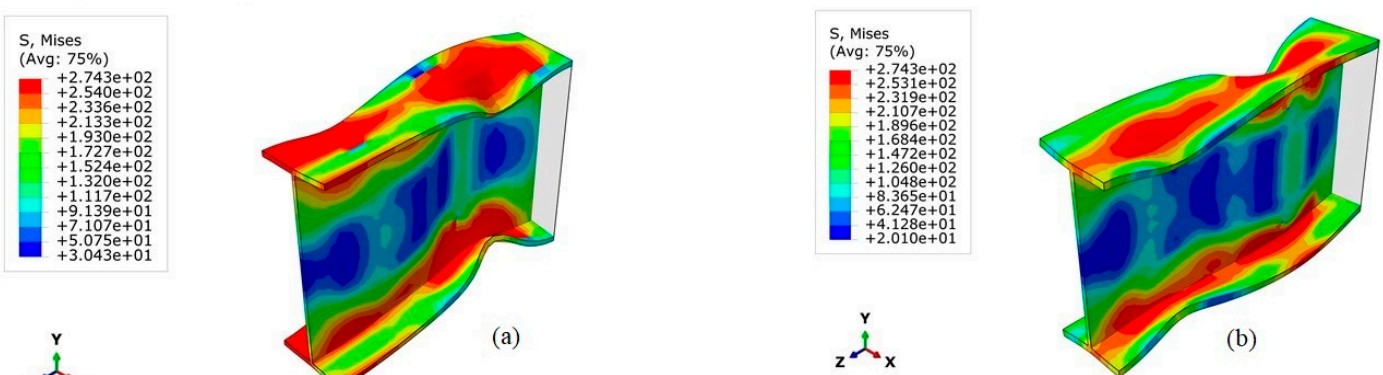

**Figure 10.** Contour plots at 500 °C for (**a**) W14 x 22 with $L/r$ = 20 and (**b**) W21x73 with $L/r$ = 20.

### 4.2.2. Flexural-Strength Assessment

Figure 11 shows the flexural-strength capacities for steel sections W14x90 and W14x22 at various slenderness ratios ($L_b/r$) ranging from 20 to 200 at a temperature of 500 °C, based on the results by Takagi and Deierlein [27] and compared against the Appendix 4 equations [12]. Instead of regenerating the entire strength curve to verify the numerical-modeling approach, a single case with a slenderness ratio of $L_b/r$ = 60 at 500 °C was compared against the provided curves. Individual member flexural responses with a laterally unbraced length ($L_b/r$ = 60) and laterally braced cases ($L_b/r$ = 0) are presented in Figure 12. As shown by the responses, the flexural-capacity ratios for the members of

W14x22 and W14x90 were obtained as 0.52 and 0.57, respectively. These results correspond well with the curves shown in Figure 11 for the slenderness ratio of $L_b/r$ = 60.

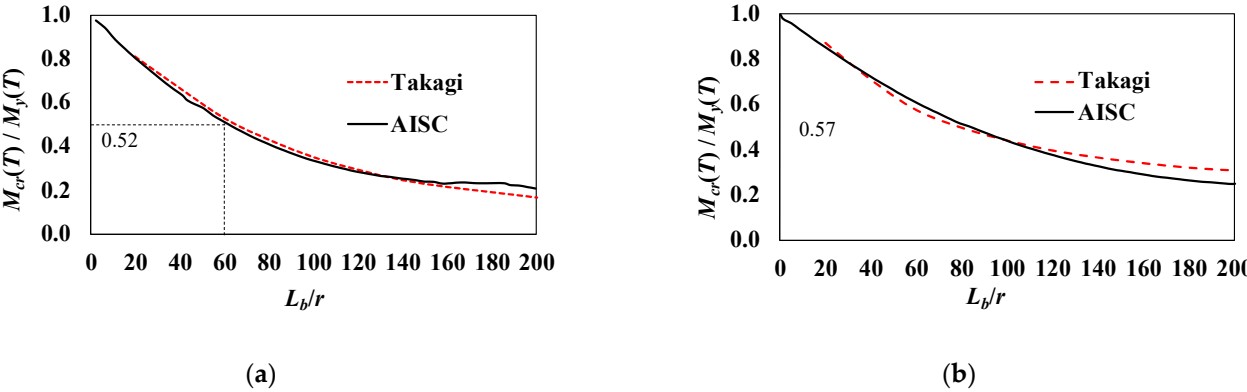

**Figure 11.** Flexural-strength curves obtained from [12,27] at 500 °C: (**a**) W14x22; (**b**) W14x90.

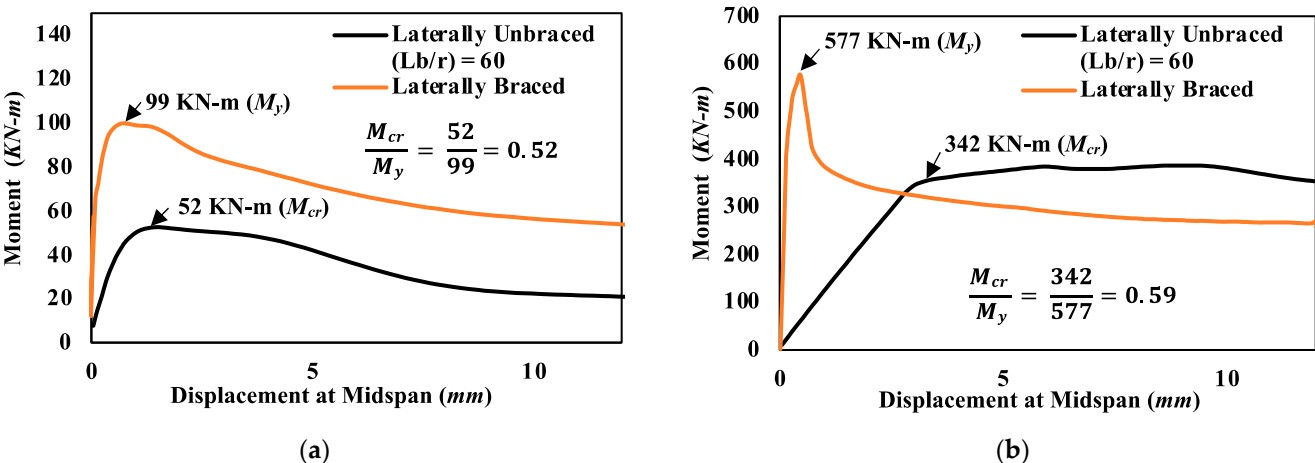

**Figure 12.** Flexural-response comparisons for $L_b/r$ = 60 and $L_b/r$ = 0 at 500 °C: (**a**) W14x22; (**b**) W14x90.

### 4.2.3. Beam-Column Strength Assessment

To assess the beam-column strength at elevated temperature, a final member study was conducted to investigate the combined axial load and bending-moment loading case. The study focused on a lower bound-slenderness ratio of $L/r$ = 20 to highlight the impact of low slenderness ratios on the interaction curves and to investigate the applicability of existing design equations to cases where local buckling is dominant in the member response. The results shown in Figure 13 present the axial-force and bending-moment interaction-curve comparisons for a W21x73 section at 20 °C, 200 °C, 400 °C, and 600 °C. The FEM results were compared against the Appendix 4 equations [12] for a slenderness ratio of $L/r$ = 20. The M-N-T interaction curves showed good agreement with the current AISC equations for low-temperature cases (20 °C, 200 °C); however, the results deviate for the high-temperature cases of 400 °C and 600 °C, particularly for the high-axial-force and low-bending-moment load cases. This discrepancy in the curves was expected from the results of the compression-only cases, as discussed in Section 4.2.1, since the Appendix 4 provisions do not take into consideration the local buckling as a limiting factor in the compressive-strength calculations. This study confirms that utilizing the AISC equations for calculating the strength of beam-column members at elevated temperatures yields unconservative estimations for members with a slenderness element (e.g., noncompact web) at low member-slenderness ratios ($L/r$), particularly when less than 60.

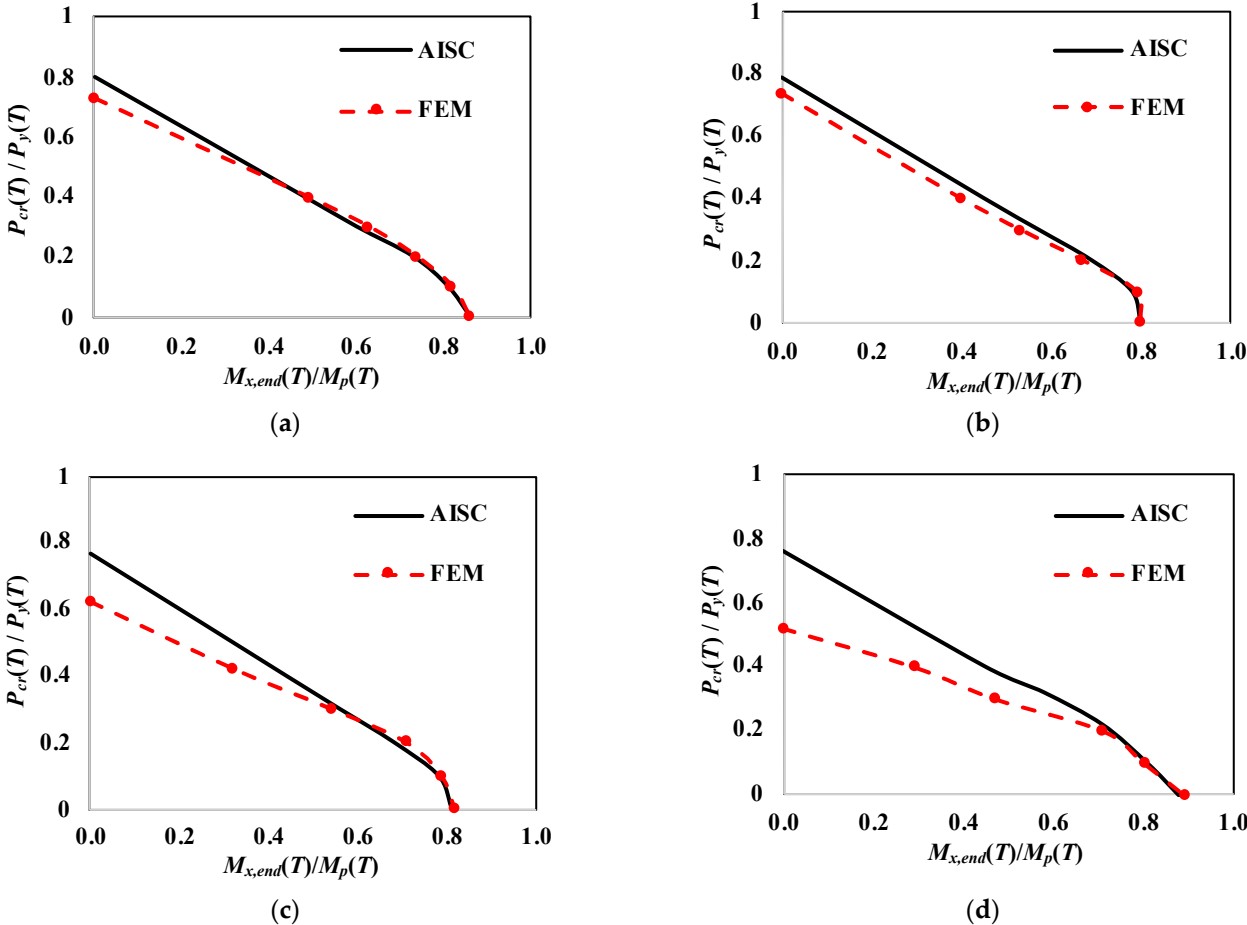

**Figure 13.** Beam-column strength for W21x73 with $L/r$ = 20: (**a**) 20 °C; (**b**) 200 °C; (**c**) 400 °C; (**d**) 600 °C.

4.2.4. Moment-Connection Capacity Assessment

The developed FEM methodology was employed to assess the capacity of a beam-column moment connection at elevated temperatures. The welded unreinforced flange-bolted web (WUF-B) moment connection was selected for the investigation as a commonly researched prequalified steel moment connection available in AISC 358 [28]. A beam-column subassembly assumed to be a part of a steel moment-frame structure with the WUF-B connection was used to conduct the connection-strength study. The subassembly member dimensions were adopted from a prior NIST study [10] that developed prototype steel-frame buildings representative of typical US construction practices. The subassembly consisted of a 1.5 m-long single cantilevered W21x73 beam attached to a W18x119 column that was 4128 mm in height using a WUF-B connection, as shown in Figure 14. The WUF-B connection consisted of a 13 mm × 305 mm × 152 mm shear tab bolted to the beam web on one end using three 25 mm-diameter high-strength bolts and welded to the column flange on the other end using an 8 mm fillet weld on both sides of the shear tab. The beam flanges were connected to the column flange using complete-joint-penetration (CJP) groove welds with backing bars. Continuity plates were provided to prevent any local flange or web deformations in the column. The structural steel members (beam and column) were made of ASTM A992 steel with a nominal yield strength of 345 MPa, and the shear tabs and continuity plates at the beam-column connections were from ASTM A36 steel with a nominal yield strength of 250 MPa. Bolts used for connecting the beam web to the shear tab were ASTM A490 with an ultimate strength of 896 MPa. The subassembly was loaded at the free end of the beam, and web-stiffener plates were attached near the loaded end to minimize local deformations.

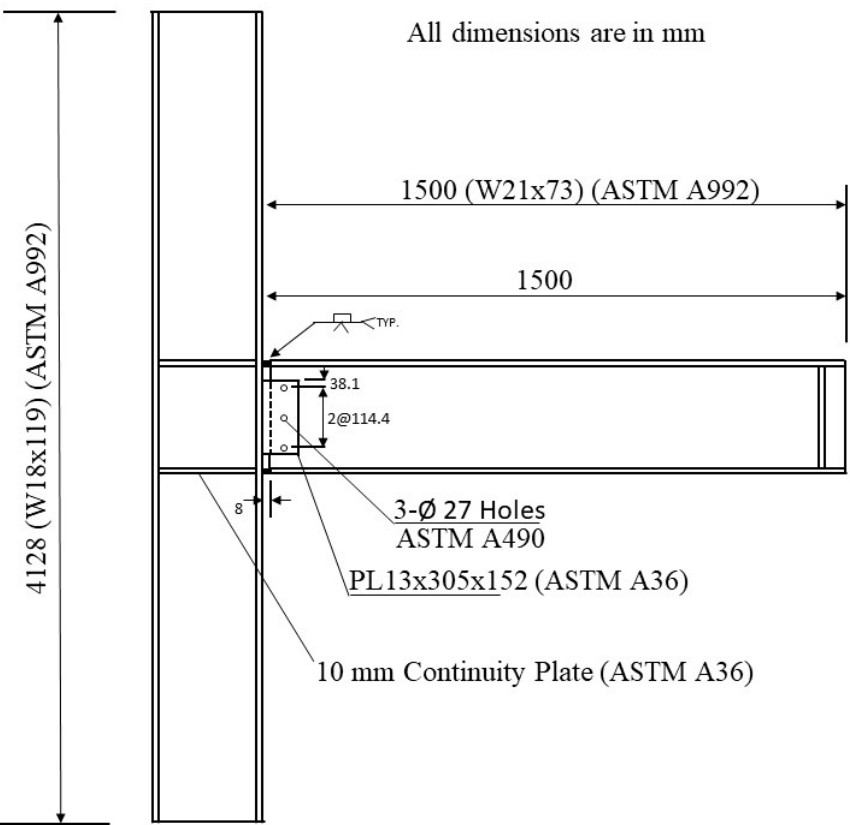

**Figure 14.** Subassembly details to study moment connections, adapted from [10].

The subassembly was subjected to uniform temperatures and the response was obtained for temperatures of 20 °C, 200 °C, 400 °C, and 600 °C through computational analysis. The analysis was strictly limited to the subassembly response at elevated temperatures and the thermal expansion during heating was neglected by assuming that the system was thermally unrestrained. Similar to the member-strength studies, the connection response at elevated temperatures was studied by subjecting it to (1) compressive-only load, (2) flexure-only load, and (3) combined axial-force and bending-moment cases. Additionally, the influence of (i) NIST [15] and Eurocode [16] material models and (ii) including material degradation and fracture models on the connection response was studied. The interaction curves obtained for the moment connection subjected to the combined load case were compared against the beam-column strength equations provided in Appendix 4 (Equations (1) through (5)) due to the lack of connection-strength equations for elevated temperatures.

The load-displacement responses for the compression-only load case are presented in Figure 15a. In the presented curves, the compressive load response against the horizontal displacement at the free end was plotted for various temperatures of 20 °C, 200 °C, 400 °C, and 600 °C using both the NIST and Eurocode stress–strain models. The horizontal lines in the figure indicate the beam axial strength (squash-load capacities) at elevated temperatures calculated based on the gross area and temperature-modified yield strength, $A_g \cdot F_y(T)$. The results indicate that the analyses using the NIST material models exceeded the squash-load capacity at large displacements. The Eurocode material models, on the other hand, provide a lower bound response with relatively lower stiffness and strength. The results also indicate that local web or flange buckling was not a limiting factor in the strength since the member-slenderness ratio of the beam was relatively high ($L/r = 36.2$ using $K = 2.0$) and controlled by global buckling (i.e., flexural buckling)-limit states.

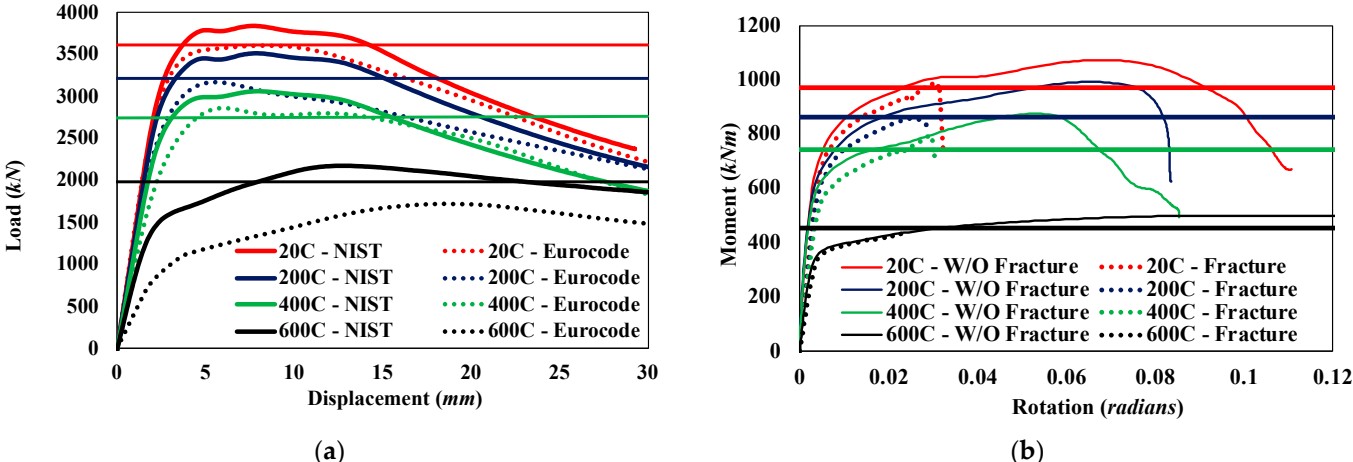

**Figure 15.** Subassembly-response curves for (**a**) compression-only and (**b**) flexure-only cases.

The moment-versus-rotation responses for the flexure-only cases at temperatures of 20 °C, 200 °C, 400 °C, and 600 °C are presented in Figure 15b. These models were also analyzed by incorporating fracture models for steel members while considering only the NIST material models (the Eurocode results are presented in the author's thesis [29]). To define the material degradation, a damage model was used to simulate ductile fracture in steel after reaching a strain threshold that corresponded to damage initiation. Based on previous literature and examining reported stress vs. strain curves, the damage-initiation strain was specified at 5% and the triaxiality for uniaxial loading was specified to be 1/3 [30]. An element-deletion criterion was enforced when an ultimate strain threshold of 10% was reached. For the models incorporating the fracture model, fracture was generally initiated in the shear tab and followed by fractures in the flange region that eventually led to failure in the connection.

The horizontal lines in the figure indicate the flexural capacities at different temperatures for the beam and correspond to the temperature-modified plastic-moment capacities, $Z_x \cdot F_y(T)$. The figure also highlights the influence of utilizing the fracture model in the analysis results of the flexural members as a marginal strength reduction between the two models. The peak strength difference due to including the fracture model was calculated to be up to 16%. The reduction in the deformation and rotation capacities were more substantial in the models accounting for fracture, as the peak displacement and rotation was reduced by 77% and 72%, respectively. It should be noted that the damage model was not temperature dependent and was considered to be a stringent criterion for the temperatures studied in this investigation.

Consistent with the previous analysis results, the flexural-response curves indicate that the finite-element models analyzed using the Eurocode stress–strain model resulted in about 10–15% less capacity than the NIST material models up to temperatures of 400 °C, with a substantial drop in stiffness and capacity for temperatures beyond 400 °C. This discrepancy was expected as a result of Eurocode 3 models accounting for rate-dependent effects such as thermal creep when defining the stress–strain relationship at various temperatures (as discussed in detail in Section 3.1). In contrast, the NIST stress–strain relationship did not account for any rate-dependent effects at elevated temperatures and resulted in a higher yield stress and more prevalent strain-hardening response.

The subassembly model was finally subjected to combinations of axial-force and bending-moment demands to obtain the connection capacity under combined loading. The subassembly models were subjected to increasing moments under a constant level of compressive force. The moment during failure was recorded as the failure moment associated with the applied compressive force. Figure 16 presents the axial force-bending moment-interaction curves at elevated temperature (M-N-T) obtained using the NIST and Eurocode 3 material models, which are also compared against the AISC Appendix 4

beam-column strength curves. The curves obtained from the NIST and Eurocode 3 models indicate good agreement with the interaction curves generated by the AISC beam-column member strengths. Up to 400 °C, the AISC curves did not indicate a significant change in the moment capacities, with a slight reduction in the compressive strength. This was expected, as the yield-strength proportionality limit did not decrease up to 400 °C (see Figure 3). However, at 600 °C, a considerable drop in the capacities was observed due to the significant reduction in the proportionality limits.

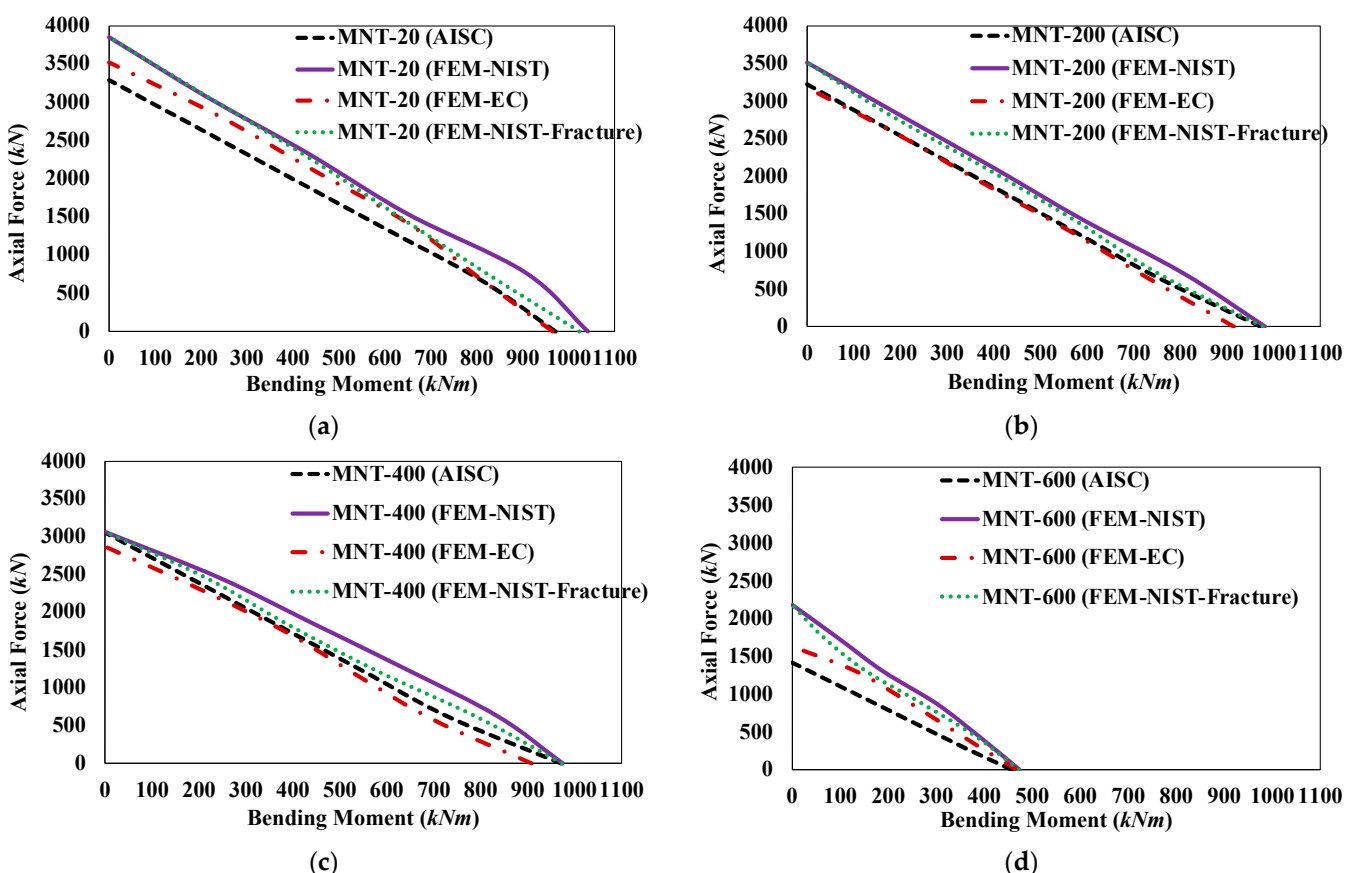

**Figure 16.** Comparison of the M-N-T interaction curves between FEM results and AISC [12]: (**a**) 20 °C; (**b**) 200 °C; (**c**) 400 °C; (**d**) 600 °C.

The AISC member-strength equations, although originally developed for members, showed reasonable agreement compared to the axial force-bending moment-interaction capacity curves of the moment-connection subassembly with members having slender elements for compression, such as the W21x73 beam section. The agreement between the analysis results and interaction curves is credited to having the subassembly-failure modes largely driven by member failures near the connection but not in the moment connection; therefore, the beam-failure modes dominated the subassembly behavior. In addition, the beam included in the subassembly study had a high slenderness ratio ($L/r$ = 36.2 and $K$ > 2.0) that was in excess of the slenderness limit to observe any local buckling-failure modes ($L_c/r$ = 60).

## 5. Discussion

The research presented in this paper focused on the structural behavior of members and connection-moment frames at elevated temperatures resulting from compartment fire. Three-dimensional finite-element numerical models were employed to evaluate the structural response of beams and connections under uniform temperatures. To verify the modeling approach, benchmark numerical models were developed and compared against

prior experimental results of beam-column subassemblies with moment connections. The analysis results accurately predicted the test results and failure modes reported in the experimental studies at different temperature levels. The established benchmark finite-element-modeling approach was then employed for assessing the member capacities of typical floor beams (with slender elements for compression) while subjected to compression, flexure, and combined force and bending moments at elevated temperatures. The analysis results were also compared against the member-strength equations at elevated temperatures provided in AISC Appendix 4 [12]. The results indicated that the AISC equations for compression-only cases were unconservative for members with low slenderness ratios ($L/r < 60$), as this outcome was also observed in the combined-load cases.

The member-level study was followed by a subassembly study that included realistic member sizes and dimensions that were representative of steel moment-frame buildings according to US construction practices. The investigation focused on the beam-to-column moment connection to obtain the fundamental connection response under compression-only, flexure-only, and combined-load cases at different temperature levels. The analysis results indicate that the moment-connection behavior was primarily controlled by failure modes observed near the beam ends. As a result of member-controlled failure modes, the axial load and moment (M-N-T)-interaction relationships obtained using the AISC Appendix 4 member strengths showed good agreement with the subassembly response. Thus, it is recommended to use the AISC Appendix 4 [12] equations for estimating strength estimates for moment connections for member-slenderness ratios greater than 60.

The results obtained from this study contribute to furthering the understanding of member and connection behavior of a moment frame under the events of fire hazards and the influence of different material models. However, this research study has limitations and some pertinent issues have not been addressed, which could lead to an improved understanding of the behavior. Some of these future study recommendations are listed below:

- Additional investigations on the connection behavior under reversing axial loads (from compression to tension) that represent the heating phase followed by the cooling phase.
- The effects of thermal stresses due to expansion (or contraction) on the moment-connection response should be investigated with more realistic modeling of thermal and restraint effects.
- Steel members without a composite slab were considered in this study; therefore, investigations that include the effects of concrete slabs are recommended.

## 6. Conclusions

This paper presents a computational study evaluating the structural response of moment frames at elevated-temperature situations resulting from fire events. Fire is a hazardous event that subjects the structure to a distinctive set of demands during its performance to resist substantial forces and excessive deflections or rotations that may lead to extensive damage. US codes and standards provide design approaches wherein the structural floor beams and connections of moment frames are designed against demands that rely on sets of load combinations. These load combinations are typically comprised of gravity-load combinations, where thermally induced demands are often not part of the design approach. Past research on steel connections subjected to elevated temperature simulating fire events have indicated that these thermally induced demands are primarily dependent on the temperature change and connection type. This phenomenon is a particular concern for stiff connections (e.g., moment connections) that are expected to be subjected to significant stresses due to their high rotational and translational stiffness, thus becoming susceptible to various failure modes due to the combined effects of bending moments and axial loads during a fire event. Therefore, it is imperative to quantify the behavior of moment frames comprised of steel beams and moment connections against the combined axial force-bending moment interaction (M-N-T) that may occur during a

fire event. The research was conducted by benchmarked finite-element models that used temperature-dependent material properties provided by Eurocode and NIST. Based on the numerical investigation results, the following conclusions are drawn:

(1) The developed numerical models were capable of capturing the behavior and response of beams, columns, and connection elements of typical moment frames. The primary influence on the analysis results were the variations in the temperature-dependent material properties used in the models. Incorporating damage evolution and failure criteria in the material model had a relatively minor impact on the overall moment-connection strength.

(2) The Eurocode 3 material properties, which are the basis for the mechanical properties of steel at elevated temperatures used in the current AISC specification, resulted in more conservative capacity estimates in comparison to the NIST material model. This is mainly because the Eurocode 3 uses elastic-modulus and yield-strength reduction factors that are considerably smaller than those of the NIST models, particularly for temperatures above 400 °C.

(3) The analysis results indicated more accurate comparisons when NIST-developed temperature-dependent material properties were used compared to the Eurocode model. The improvement in the results is credited partly to the NIST models accounting for types of steel such as plates and bolts more accurately, and not accounting for rate-dependent effects such as thermal creep. Therefore, the NIST material models are recommended for detailed analysis of steel members at elevated temperatures.

(4) Failure modes observed for connections of moment frames with slender beam members were typically controlled by buckling of beam members occurring near the connection region and therefore governing the connection strength. Therefore, calculation procedures used for member design were found to be applicable for obtaining moment-connection capacities.

(5) The AISC Appendix 4 equations for compression design provide unconservative results for members with slender beam elements due to neglecting the local buckling-failure modes. Similarly, the beam-column strength equations used for estimating the capacity of members subjected to combined axial force and bending-moment cases in Appendix 4 provided unconservative results for typical beam members at low slenderness ratios. Therefore, the AISC equations are recommended for predicting the capacity of moment-frame members with slenderness ratios greater than 60.

**Author Contributions:** Conceptualization, S.N.C. and K.C.S.; methodology, K.C.S.; validation, S.N.C. and K.C.S.; formal analysis, S.N.C.; investigation, S.N.C. and K.C.S.; data curation, S.N.C. and K.C.S.; writing—original draft preparation, S.N.C.; writing—review and editing, K.C.S.; visualization, S.N.C.; supervision, K.C.S.; project administration, K.C.S. All authors have read and agreed to the published version of the manuscript.

**Funding:** This research received no external funding.

**Institutional Review Board Statement:** Not applicable.

**Informed Consent Statement:** Not applicable.

**Data Availability Statement:** Not applicable.

**Acknowledgments:** The authors would like to acknowledge the support of Auburn University for providing the computational resources to conduct the presented research.

**Conflicts of Interest:** The authors declare no conflict of interest.

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
