# Peer review of "Behavior and Capacity of Moment-Frame Members and Connections during Fire"

_fire, doi:10.3390/fire6020078_

Round 1

Reviewer 1 Report

The article shows an extensive study on the behavior of frame members and connections under fire situation.

The authors suitably approach other experimental research as reference to elaborate and adjust numerical models. 

The figures are clearly presented and help to understand the applied methodology as well as the results. 

The results obtained were properly commented. The overall mechanical behavior analysis was well established and compared with references from experimental research and the main normative codes.

Author Response

We thank the reviewer for taking the time to review the manuscript. We also believe that addressing other reviewers’ comments has improved the quality of this paper. We hope that you will find the revised manuscript to your satisfaction.

Reviewer 2 Report

The authors performed a series of numerical analyses of moment frame connections, beams, columns, and beam-column subassemblages subjected to combined gravity and fire demands. The paper has potential for scientific merit; however, needs major revisions and I do not recommend this paper for publication. The following are suggestions for improvement for this paper and study. 

1. The manuscript is poorly written with many subject-verb agreement issues, varying tenses of words, redundant writing, and long sentences. I strongly suggest carefully proofreading this paper prior to resubmission. I have aimed to point out a few of the specific issues; however, they are throughout the full manuscript.

2. The authors tend to use the word "significant" to describe a qualitative evaluation of the impact of certain variables on the behavior of steel structures in fire. This word means that the relationship between two variables can be quantified. I suggest changing the word "significant" to "substantial" throughout the paper or another word of the authors' choosing.

3. Line 30: "lateral loads" are the authors implying that lateral loads are applied during a fire? The authors acknowledge the induced thermal loads, but I am not quite sure what is meant by "lateral loads" here other than earthquake and wind forces. Buildings are not designed to resist lateral loads at the same time as a fire demand.

4. Line 31: change "requiring satisfactory performance concerning resilience" to "required resilience performance criteria ..."

5. Lines 32-34: "Available prescriptive design .... ": Prescriptive design does not have load combinations and often load carrying capacity is not the failure criteria of assemblies with a prescriptive design fire resistance rating.

6. Line 34: "past recent events": this is redundant. Change to "past events"

7. Lines 36-37: remove "... that subsequently led to various fire events following the seismic activities." this is redundant since the authors already stated compartment fires started after earthquakes.

8. Lines 34-38: there are more recent post-earthquake fire examples within the last 20 years. I might suggest the authors add those to the references. For instance the 2011 Tohuku Earthquake and Tsunami and the 2019 Ridgecrest Earthquake.

9. Line 38: "Past events ..." please specify what event you are talking about. In the paragraph it seems you are only talking about fires whereas in the previous paragraph the authors were talking about post-earthquake fires.

10. Line 42: " ... delaying the collapse as a prescriptive method." First, please indicate what the collapse is referring to. Second, the prescriptive fire protection design method is not for preventing collapse, but rather for life safety. Please modify the sentence to reflect the correct performance objectives for prescriptive fire safety design. I might suggest referring to the IBC for this.

11. Line 43: remove the word "providing"

12. Line 45: Please add a citation that designing a structure to resist fire demands using performance-based design is more efficient and economical. There are a few academic articles on this.

13. Lines 46-70: the authors spend a paragraph describing the performance of gravity connections; however, this is not the focus of their research. This topic has been researched extensively for over four decades and the authors cite three papers. I don't see the relevance of this paragraph to the paper and the authors do not describe the behavior correctly (as seen from subsequent comments here). I suggest removing this paragraph as it distracts from the focus of the paper.

14. Lines 52-53: "and the forces generated through floor beams ..." the forces are generated when the rotation and deformations are restricted in the gravity system. It is the restraint of the cooler bays and the restriction of rotation of the connections that induce these forces.

15. Line 54: "Due to the relative size and limited exposure, ..." it is only due to the limited exposure since the connection is at the beam-column interface.

16. Lines 60-62: remove the sentence that begins with "Simple (shear) connections typically used in gravity steel frame ..." because all of that information has already been stated.

17. Line 65: ".... thermal stresses near the beam capacity." I am not quite sure what the authors mean by this. I might suggest rephrasing this to be a bit clearer on the point that is trying to be made. 

18. Lines 78-80: remove " ... that consisted of a beam supported on columns at each end with moment connections." as this is just the definition of a moment frame.

19. Line 84: " ... with varying beam temperatures." does this mean a singular set temperature or the beam had a thermal gradient that changed with time?

20. Line 85: what is "temperature cycling"? This comment is related to the previous comment. I am not quite sure what the thermal exposure was.

21. Line 86: remove "at service-load condition" the authors already stated gravity loads were applied to the beam.

22. Line 87: Change "Figures 1" to "Figure 1" 

23. Line 92: change "were achieved" to "was achieved". In addition, I recommend making this sentence two sentences. It is a bit long and I am not quite sure what the authors are referring to with being achieved. 

24. Figure 1: What is the positive and negative with respect to compression and tension? Please add this to the discussion.

25. Section 2: The objectives and approach could use some organization and refinement. The objectives were not consolidated into a clear paragraph, rather scattered throughout this section. I would recommend them all being listed up front with the overall objective of the study. 

26. The authors only state they used structural analysis models; however, there were a few studies simulated in transient state and therefore, shouldn't the authors have to perform a heat transfer analysis? I specifically point these out in subsequent comments.

27. Line 122: provide a citation for the design codes the authors are referring to.

28. Line 128: The authors state that the benchmarked models were benchmarked against moment-rotation data; however, none are reported in the next section. Either remove or add that comparison in.

29. Line 129: The authors state that the benchmarked models were benchmarked against failure modes; however, none are reported in the next section. Rather only deformation of the column section. Either remove or add that comparison in.

30. Line 136: remove discussion of comparing the results of the analysis against member strength equations as that was already stated in this section.

31. Line 144: change "materials" to "material"

32. Lines 162-163: The authors do not explicitly model the weld in any of their models and do not compare any of the axial demands against weld strength. Later in the evaluation of the subassembly, the authors use an all welded connection. The authors should justify this here or at least acknowledge the limitation of their study as they are not considering failure or damage within the connection.

33. Section 3: How was pre-tension of bolts modeled?

34. Lines 166-167: remove "so the contact was defined using tie constraints as it was rigidly attached to the surface of the column" this was already stated in Lines 162-163.

35. Lines 186-187: This statement conflicts with the plots shown in Figure 4 and the following discussion of stiffness.

36. Line 204: Eurocode 3 provides different retention factors for different types of steel. The statement that Eurocode 3 does not have different steel properties at elevated temperature for bolts and other members is incorrect. Please revise this statement. 

37. Figure 3 title: the way this title is written implies that Eurocode 3 states the AISC Specification reduction factors, when it is actually the opposite. I might suggest just citing the AISC Specification and Eurocode 3 rather than stating "used in AISC Specification according to Eurocode 3".

38. Lines 245-246: The authors state there was loss of pretensioning in the bolts, but don't state how this was simulated.

39. Section 3.2.1: How were the temperatures obtained for the transient test? If through a heat transfer model, please describe the modeling methodology. If through another mechanism, please describe.

40. Figure 5 caption: Please state which specimen is shown in (a). There seems to be an (a), (b), and (c) all in (a). Should this be two figures? In addition, can the photograph be utilized within this paper without permissions?

41. Section 3.2.1 and Section 3.2.2: the authors state there is close alignment between the FEM model and the experimental data multiple times. This could be strengthened through an error calculation. Please provide some quantification of this closeness to provide context. This could be broken up by steady state versus transient analysis and Eurocode and NIST material properties.

42. Section 3.2.1 is the only evaluation of the NIST versus Eurocode material properties. The authors then only use the NIST material model in Section 3.2.2 and then switch back to Eurocode for the final analysis. However, one of the objectives is to evaluate the applicability of these two models. One analysis is not enough to evaluate that. In addition, the authors state they use the NIST model in Section 3.2.2 because it performed better in 3.2.1 than the Eurocode model, but then switch back to Eurocode in the final analysis of the subassemblies and beam/column. I suggest the authors think about this objective and either use both models throughout or one consistent model and remove this as an objective.

43. Line 294: "... were chosen for develop benchmarking the finite element ..." please revise this phrasing to "were chosen to develop a benchmarked finite element ...."

44. Lines 312-318: The authors use the phrases "close agreement", "correlated well", "captured well" without any quantification of how well the FE model performed. There are many different methods of performing error calculations for fire and for other structural modeling - comparing the area under the results curve, straight error calculations, etc. I think this would strengthen the work the authors are presenting.

45. Section 3.2.2: how did the authors perform the heat transfer analysis for the transient test?

46. Figure 6: The authors should think about revising Figures 6a and 6b to have less heavy lineweights so that the individual lines are able to be read.

47. Section 4: The authors perform benchmarking analysis for connections and then focus their parametric study on members and a connection that only uses tie constraints for welds. I am not quite seeing how Section 3 and 4 tie together. 

48. Section 4.1: The equations in AISC Specification Appendix 4 are for gravity frames; however, the authors seem to be applying these to moment frames. Please provide justification.

49. Lines 343-344: There is only one equation for column buckling in Appendix 4 and there are two in Chapter E. These are not the same because in fire we don't have elastic buckling so there is only an inelastic buckling equation.

50. Line 365: Shouldn't the authors be referencing interaction equations in AISC 341 rather than 360? 

51. Section 4.2: I was a bit confused why the authors use the Takagi and Deierlein study for their evaluation. First, this was for all gravity columns and beams and the authors are supposed to be studying moment frames. Second, a W14x22 steel section is quite small for a moment frame. 

52. Lines 378-380: Remove sentence: "The final case was conducted to obtain interaction equations ..." as this has been stated multiple times throughout the paper.

53. Lines 381-383: It is not clear what is new from the authors study that hasn't been already done by Takagi and Deierlein. I might suggest that the authors revise this portion to make that a bit clearer.

54. Lines 389-391: "The nominal strength properties at ambient and elevated temperatures ..." Please revise this sentence for clarity. It reads as though temperature-dependent material properties were not considered and I don't think that is the intention of the authors.

55. Line 392: "In these member studies, simply supported boundary conditions ..." This is confusing to me since the authors aimed to look at moment frames. 

56. Section 4.2.1: The authors conclude that members with low slenderness ratios cannot have their capacity predicted by the AISC Specification Appendix 4 equations; however, these would correspond to short columns. The ry for a W14x22 is 1.04, so Lc/r = 20 would be a approximately 20 inch tall column. An Lc/r < 50 would be a column with a height less than about 50 inches. This is unrealistic. What is the applicability of what the authors are concluding, particularly for moment frame columns? There are substantial amounts of research on steel columns in fire, I might suggest that the authors look into this research a bit further to examine what previous researchers have explored on this topic. 

57. Figure 11 states this is showing flexural strength; however, the y-axis is the critical buckling axial force divided by yield axial force. 

58. Please update Figure 11b as it shows up greyed out and the lines showing the 0.57 lines are not visible.

59. Figure 12: please update the figure title to reflect what is being plotted. These are force-deformation curves. Is this the deformation at midspan?

60. Figure 13: There are two (a)s and (b)s in the paper - please revise

Author Response

We thank the reviewer for the detailed review and for providing extensive comments. We’ve tried our best to address the review comments by making a major revision to the manuscript, which we believe improved the quality of this paper substantially. We hope that you will find the revised manuscript to your satisfaction. Please see the attached file for the responses to each comment.

Reviewer 3 Report

The paper presents the broad study focused on the structural behaviour of members and connections of moment frames exposed to elevated temperatures.

The issue of steel frames exposed to fire is not new in the field of steel frames however, the investigation of the effect of elevated temperatures on behaviour of steel frame members is focused mainly on members exposed only to bending. In this study authors investigate the structural behaviour and capacity of moment frames members and connections subjected to combined effects of bending moment and thermally-induced axial force.

Below are my questions and comments to the manuscript:

The writing logic is a bit chaotic. The aim of the work (a study of floor beams), emphasized in almost every chapter, indicates that the text will concern the creation of M-N-T interaction curves for these beams, while the description of member strength assessment at elevated temperatures takes up 4 pages (out of 20 pages), and the next 4 pages are taken up by the description of beam-column strength assessment.

Perhaps a good solution would be to shorten section 2 (Objective and Approach) and combine it with the Introduction. For example lines 105 to 113 are repeated later as lines 130 to 141. Also, for the most part, Section 5 is a repetition of Section 2 and should be rewritten.

It may also be worth considering dividing the paper into two, one of which would concern strength assessment of beams and the other connections of steel frames exposed to elevated temperatures.

Were the FEM models described in Section 3 used for the analysis described in Section 4, or a new model for assessment the members and connections strength created? If so, what was the purpose of the benchmarks described in chapter 3?

Figures should be more formal for improving the level of this manuscript.

*Introduction:

1) The cited references are relevant however there are also many recent publications regarding beam to column connections in fire.

2) After summarizing the research work of others, the authors should put forward the main conclusions from the literature review which inspired the authors of the work to undertake such a research topic.

* Objective and Approach

1)   Line 120 to 121: were the M-N-T curves developed for the joints compared with the curves created on the basis of equations developed for steel members?

*Member Strength Assessment at Elevated Temperatures

1)   Figure 9: in the caption of the figures there are two letters a) and two letters b) instead of: a, b, c, d

2)   Figure 10: should be: a) and b) instead of c) and d)

*Beam-Column Strength Assessment

1)   how was the loading of the connection carried out? how the interaction curves were created?

2)   Figure 13: in the caption of the figures there are two letters a) and two letters b) instead of: a, b, c, d

3)   Figure 16: as above

*Conclusions

1)   Conclusion no.3: please note that Eurocode 3 also gives values of reduction factors for bolts.

Analyses and benchmarks presented and described in this paper have shown that the results obtained using material parameters developed in accordance with NIST lead to overestimation of joint load capacity, hence whether they are definitely recommended for use?

2)   Did the authors of the work consider modifying existing AISC equations or developing new equations for predicting the capacity of moment frame members with slenderness less than 60?

Author Response

(The authors gave the same response as above.)

Reviewer 4 Report

The article can be published in its current form.

Author Response

(The authors gave the same response as above.)

Round 2

Reviewer 2 Report

The authors address the majority of the comments provided by the reviewers and the reviewer thanks the authors for their efforts. Please consider the following follow up comments to continue to improve the manuscript:

1. The authors are still using the word "significant" throughout the manuscript in an inappropriate manner. Please thoroughly review the manuscript for this word.

2. Lines 66 - 70: the reviewer agrees there might be larger axial forces or moment demands in moment connections; however, on the other hand these connections are actually designed for high moment and axial force demands. Therefore, it is unknown whether these demands are higher than the capacities they are designed for. I suggest adding this into the sentence.

3. Line 86: I am still a bit confused about the loading demands on the members due to the phrasing "service-load conditions". Usually this terminology is meant for deflections. Do the authors mean using the load combination 1.2D + 0.5L or actual service load conditions of Dead + Live? The first is a load combination.

4. Line 108: The authors are missing a citation. The citation is listed as "[X]"

5. Lines 275-276: The authors state there is an increased accuracy between using the NIST stress-strain material properties than using the Eurocode stress-strain material properties, but do not quantify this. It is not clear how the authors determined this accuracy if there is no quantification other than visual observations between the two curves. I do strongly suggest that the authors perform an error analysis. This is often performed by researchers throughout the literature. The same comment applies to Lines 283 - 292. I understand the authors are hesitant against this from their reply to the last comment on this; however, it is not an unreasonable ask as there seems to be plenty of data for the authors to compare against.

Reviewer 3 Report

The authors have satisfactorily addressed and answered the reviewer comments and questions.

Author Response

We once more thank the reviewer for taking the time to review the manuscript. We also believe that addressing other reviewers’ comments has improved the quality of this paper. We hope that you will find the revised manuscript to your satisfaction.